# Personalized Denoising Implicit Feedback for Robust Recommender System

## Abstract

While implicit feedback is foundational to modern recommender systems, factors such as human error, uncertainty, and ambiguity in user behavior inevitably introduce significant noise into this feedback, adversely affecting the accuracy and robustness of recommendations. To address this issue, existing methods typically aim to reduce the training weight of noisy feedback or discard it entirely, based on the observation that noisy interactions often exhibit higher losses in the *overall loss distribution*. However, we identify two key issues: (1) there is a significant overlap between normal and noisy interactions in the overall loss distribution, and (2) this overlap becomes even more pronounced when transitioning from pointwise loss functions (e.g., BCE loss) to pairwise loss functions (e.g., BPR loss). This overlap leads traditional methods to misclassify noisy interactions as normal, and vice versa. To tackle these challenges, we further investigate the loss overlap and find that *for a given user, there is a clear distinction between normal and noisy interactions in the user's personal loss distribution.* Based on this insight, we propose a resampling strategy to **D**enoise using the user's **P**ersonal **L**oss distribution, named **PLD**, which aims to reduce the probability of noisy interactions being optimized. Specifically, during each optimization iteration, we create a candidate item pool for each user and resample the items from this pool based on the user's personal loss distribution, prioritizing normal interactions. Additionally, we conduct a theoretical analysis to validate PLD's effectiveness and suggest ways to further enhance its performance. Extensive experiments conducted on three datasets with varying noise ratios demonstrate PLD's efficacy and robustness.

## CCS Concepts

• **Information systems** → **Recommender systems**; • **Security and privacy** → **Social network security and privacy**.

## Keywords

Robust Recommender System, Denoising Recommendation, Implicit Feedback

**ACM Reference Format:**
Anonymous Author(s). 2025. Personalized Denoising Implicit Feedback for Robust Recommender System. In *Proceedings of The 2025 ACM Web Conference (WWW'25)*. ACM, New York, NY, USA, 11 pages. https://doi.org/XXXXXXX.XXXXXXX

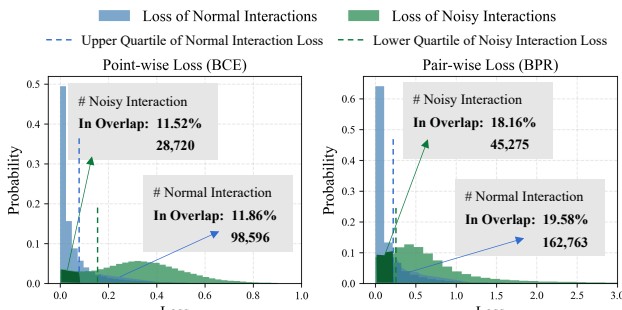

**Figure 1: Probability distribution of losses. The overlap region includes interactions that deviate from the common assumption in existing methods, i.e., where noisy interactions exhibit lower losses or normal interactions exhibit higher losses. Quantiles are used instead of max-min values to mitigate the influence of extreme values when determining the overlap region.**

## 1 INTRODUCTION

Recommender systems have become essential tools for mitigating information overload in the modern era [6, 11]. Since obtaining explicit user feedback (e.g., ratings) is often hindered by the need for active user participation, these systems typically rely on implicit feedback to capture user behavior patterns, thereby facilitating effective recommendations [4, 8, 20]. Nonetheless, factors such as human error, uncertainty, and ambiguity in user behavior inevitably introduce significant noise into this feedback [23, 40]. This noise can bias learned behavior patterns, undermine system robustness, and degrade recommendation performance [34, 40].

Mainstream methods for noise elimination in recommender systems primarily focus on reweighting feedback. A commonly observed pattern in the overall loss distribution, which represents the losses of all interactions (i.e., feedback), is that **noisy interactions tend to exhibit higher losses during training** [5, 8, 15, 26]. Based on this observation, these methods either reduce the training weight of high-loss interactions or discard them entirely. For instance, R-CE [26] assigns weights to interactions based on their loss magnitude, with higher losses receiving smaller weights. T-CE [26] proportionally discards interactions with the highest losses at a predefined rate. These methods typically compute the loss for each interaction using pointwise loss functions, such as Binary Cross Entropy (BCE) loss [8, 26].

However, we identify two key limitations in this approach. To illustrate, we separate the overall loss distribution into normal and noisy interaction loss distributions. For clarity, we define an overlap region that includes interactions deviating from the assumptions of existing methods, i.e., where noisy interactions exhibit lower losses or normal interactions exhibit higher losses:

- We observe a significant overlap between normal and noisy interactions in the overall loss distribution, as shown on the left

side of Figure 1. In LightGCN [6], trained with BCE loss on the MIND [31] dataset with a 30% noise ratio, 11.86% of normal interactions (98,596) fall within the overlap, while 11.52% of noisy interactions (35,926) also fall within this overlap.

- Moreover, this overlap becomes more pronounced when transitioning from pointwise loss functions to pairwise loss functions, such as BPR loss [18], as shown on the right side of Figure 1. Specifically, the percentage of normal interactions in the overlap increases to 19.58% (162,763, a 65.1% increase), while the percentage of noisy interactions rises to 18.16% (60,170, a 67.5% increase).

It is important to note that interactions in the overlap region cannot be reliably identified using the overall loss distribution alone. This limits the effectiveness of methods that rely solely on the overall loss distribution for denoising.

To address these issues, we investigate the causes of the overlap between noisy and normal interactions in the overall loss distribution. We find that, due to the variance in users' personal loss distributions, the losses of normal interactions for some users overlap with those of noisy interactions for others, leading to significant overlap in the **overall loss distribution**. Furthermore, we observe that, for a given user, **there is a clear distinction between normal and noisy interactions in the user's personal loss distribution**.

Based on this insight, denoising from the perspective of a user's personal loss distribution, rather than the overall loss distribution, yields more effective results. However, the variance in users' personal loss distributions and the differing ratios of noisy interactions across users make it challenging to set an appropriate drop rate for filtering noisy interactions or to adjust their weights based on interaction losses, as traditional denoising approaches do [8, 26].

Given these considerations, we propose a resampling strategy for **D**enoising based on users' **P**ersonal **L**oss distributions, named **PLD**. PLD reduces the probability of noisy interactions being optimized by resampling training interactions. Specifically, PLD first uniformly samples a user's interacted items to construct candidate pools, ensuring the stability of subsequent resampling. In the resampling stage, it selects an item for optimization from the candidate pool based on the user's personal loss distribution, prioritizing normal interactions. Additionally, we conduct an in-depth theoretical analysis of PLD, demonstrating its effectiveness and suggesting that adjusting the sharpness of the resampling distribution using a scaling coefficient can further improve the probability of sampling normal interactions. Extensive experiments show that PLD achieves state-of-the-art performance across various noise ratios, not only with BCE loss but also with BPR loss.

The main contributions of our work are as follows:

- We identify the limitations of existing loss-based denoising methods, highlighting the significant overlap between normal and noisy interactions in the overall loss distribution.
- We find that, for a given user, there is a clear distinction between normal and noisy interactions in the user's personal loss distribution. Leveraging this insight, we propose a resampling strategy for denoising, PLD.

- We conduct an in-depth theoretical analysis of PLD, proving its effectiveness and suggesting ways to further enhance its performance.
- Extensive experiments validate the superiority of our proposed method across various datasets and noise ratios.

## 2 RELATED WORK

This section briefly reviews the research on collaborative filtering and denoising implicit feedback.

### 2.1 Collaborative Filtering

Collaborative Filtering (CF) remains a fundamental technique in the design of recommender systems and has been extensively adopted in numerous research efforts [2, 21, 38]. At its core, CF operates on the principle that users with similar behaviors or preferences are likely to have aligned future choices, making it a powerful tool for predicting recommendations [12]. A widely used method within this paradigm is Matrix Factorization, which models latent relationships between users and items by factorizing the interaction matrix [11]. This interaction matrix can be constructed from both explicit feedback, such as ratings, and implicit feedback, which includes indirect behavioral signals like clicks, views, and purchases [10]. Although implicit feedback is often noisier and lacks clear negative signals, it provides a wealth of data that is crucial for building recommendation models in real-world scenarios where explicit ratings are scarce [10].

In recent years, the introduction of deep learning has expanded CF's capabilities, particularly for handling the complexities of implicit feedback. Neural-based models can capture more nuanced user-item interactions, often observed through implicit signals. For example, CDL [24] integrates auxiliary item data into CF using neural networks, effectively addressing data sparsity. Similarly, NCF [7] replaces the traditional dot product operation with a multi-layer neural architecture, which is better suited for modeling the intricate patterns found in implicit user interactions. More recently, the rise of Graph Neural Networks (GNNs) has inspired graph-based CF models [28, 33, 35], such as NGCF [27] and LightGCN [6], which have shown exceptional performance in leveraging implicit feedback. Despite these advancements, a persistent issue remains: the vulnerability of these models to noise, particularly from implicit data, which continues to undermine their robustness [40].

### 2.2 Denoising Implicit Feedback

Recommender systems that rely on implicit feedback have garnered substantial attention. However, recent research highlights their susceptibility to noise in implicit feedback [25, 40]. The primary strategies for mitigating noise in recommender systems can be broadly categorized into two types [40]: reweight-based approaches [5, 8, 15, 22, 26, 30, 37] and self-supervised approaches [29, 32, 36].

**Reweight-based Methods.** These approaches aim to reduce or eliminate the influence of noisy interactions by adjusting their contributions during training [8, 15, 26, 30]. Some methods reduce the weights of noisy interactions [5, 15, 26, 30], while others remove them entirely [8, 26]. A common observation driving these methods is that noisy interactions typically produce higher training losses in the overall loss distribution [5, 8, 15, 26]. For instance,

R-CE [26] leverages loss values as indicators of noise, assigning reduced weights to potentially noisy interactions, while T-CE [26] eliminates interactions with the highest loss values at a predefined drop rate. DCF [8] further addresses challenges posed by hard positive samples and the data sparsity introduced by dropping interactions. However, since normal and noisy interactions overlap in the overall loss distribution, the effectiveness of these methods is limited. Additionally, BOD [30] formulates the determination of interaction weights as a bi-level optimization problem to learn more effective denoising weights, though this approach is significantly more time-consuming.

**Self-supervised Methods.** Self-supervised approaches mitigate noise by introducing auxiliary signals through self-supervised learning [3, 16, 17, 29, 44]. For example, SGL [32] enhances the robustness of user-item representations by applying various graph augmentations, such as node dropping and edge masking. KGCL [36] incorporates external knowledge graph data to refine the masking process. Meanwhile, DeCA [29] posits that clean data samples tend to yield consistent predictions across different models and, therefore incorporates two recommendation models during training to better differentiate between clean and noisy interactions. However, self-supervised approaches rely heavily on the design of self-supervised tasks, and these heuristics cannot always guarantee effective denoising performance.

## 3 PRELIMINARY

We begin by providing a formal description of the task. Following [6, 8], we define a set of users as $\mathcal{U} = \{u\}$ and a set of items as $\mathcal{V} = \{v\}$, along with an observed interaction set, $\mathcal{I} = \{(u,v) \mid u \in \mathcal{U}, v \in \mathcal{V}\}$, where the pair $(u,v)$ indicates that user $u$ interacted with item $v$. Generally, recommendation methods based on implicit feedback are trained on interaction data, treating $(u,v) \in \mathcal{I}$ as positive samples and $(u,v) \in (\mathcal{U} \times \mathcal{V}) \setminus \mathcal{I}$ as negative samples to learn the parameters $\Theta$. The training of the recommendation model is formulated as:

$$\Theta^* = \arg\min_{\Theta} \mathcal{L}(\mathcal{U}, \mathcal{V}, \mathcal{I}),$$

where $\mathcal{L}$ denotes the recommendation loss. However, the observed interaction set $\mathcal{I}$ may contain noise, leading to a deviation between the learned parameters $\Theta^*$ and the ideal parameters $\Theta^{\text{Ideal}}$, which represent the optimal model parameters in the absence of noise. The goal of denoising methods is to mitigate the impact of noise in the observed interaction set $\mathcal{I}$ on the model parameters [39, 40, 43].

## 4 METHOD

To address the limitations of current denoising techniques [5, 8, 15, 26], we conduct a thorough investigation into the underlying causes of the overlap between normal and noisy interactions in the overall loss distribution. Then, we refine existing denoising criteria and introduce a novel resampling strategy for denoising based on users' personal loss distributions, called PLD. Furthermore, we enhance the denoising capability of PLD through rigorous theoretical analysis, resulting in a more robust and effective denoising methodology.

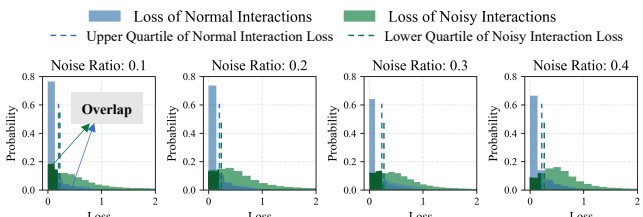

**Figure 2: Probability Distribution of losses.**

**Table 1: Statistics of overall loss distribution**

| Noise Ratio | $\|\mathcal{I}^{\mathcal{G}}_{\text{normal}}\|$ | $\|\mathcal{I}^{\mathcal{G}}_{\text{normal}}\|/\|\mathcal{I}_{\text{normal}}\|$ | $\|\mathcal{I}^{\mathcal{G}}_{\text{noise}}\|$ | $\|\mathcal{I}^{\mathcal{G}}_{\text{noise}}\|/\|\mathcal{I}_{\text{noise}}\|$ |
|---|---|---|---|---|
| 0.1 | 152,892 | 18.40% | 12,976 | 15.61% |
| 0.2 | 162,700 | 19.58% | 29,130 | 17.52% |
| 0.3 | 162,763 | 19.58% | 45,275 | 18.16% |
| 0.4 | 159,454 | 19.19% | 59,486 | 17.89% |

## 4.1 Motivation

To investigate the causes of the overlap between normal and noisy interactions in the overall loss distribution, we conduct an experimental analysis. Using the MIND dataset [31] as a case study, we introduce additional noise ratios of 10%, 20%, 30%, and 40% into user interactions and evaluate the impact on LightGCN [6][1]. Detailed description of the experimental setup can be found in Section 5.1.

To facilitate this analysis, we introduce the following notation:

- $\mathcal{I}_{\text{normal}}$: the set of normal interactions.
- $\mathcal{I}_{\text{noise}}$: the set of noisy interactions.
- $l_{u,v}$: the loss corresponding to the interaction between user $u$ and item $v$.
- $O$: the overlap region containing noisy interactions with lower losses and normal interactions with higher losses in the **overall loss distribution**, as depicted in Figure 2.
- $O_u$: the overlap region in user $u$'s **personal loss distribution**.

**Note that** quantiles are used instead of max-min values to mitigate the influence of extreme values when determining overlap regions. We further define the following sets to analyze interactions within the overlap regions:

- $\mathcal{I}^{\mathcal{G}}_{\text{normal}} = \{(u,v) \mid (u,v) \in \mathcal{I}_{\text{normal}} \wedge l_{u,v} \in O\}$: the set of normal interactions that fall within the overlap region of the overall loss distribution.
- $\mathcal{I}^{\mathcal{G}}_{\text{noise}} = \{(u,v) \mid (u,v) \in \mathcal{I}_{\text{noise}} \wedge l_{u,v} \in O\}$: the set of noisy interactions that fall within the overlap region of the overall loss distribution.
- $\mathcal{I}^{\mathcal{P}}_{\text{normal}} = \{(u,v) \mid (u,v) \in \mathcal{I}_{\text{normal}} \wedge l_{u,v} \in O_u\}$: the set of normal interactions that fall within the overlap region of the personal loss distribution.
- $\mathcal{I}^{\mathcal{P}}_{\text{noise}} = \{(u,v) \mid (u,v) \in \mathcal{I}_{\text{noise}} \wedge l_{u,v} \in O_u\}$: the set of noisy interactions that fall within the overlap region of the personal loss distribution.

**Overall Loss Distribution.** The overall loss distribution consists of the loss of all interactions. For clarity, we separate the overall loss distribution into normal and noisy interaction loss distributions. Figure 2 illustrates that normal and noisy interactions exhibit

---

[1]Similar experiments are conducted on different datasets and models, yielding consistent results. Due to space constraints, we present only the results for this configuration here.

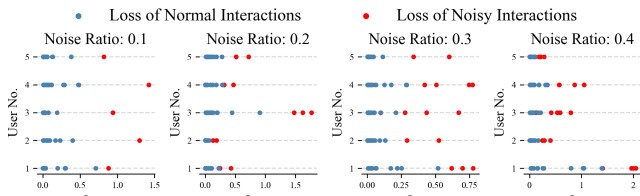

Figure 3: Personal loss distribution for five users.

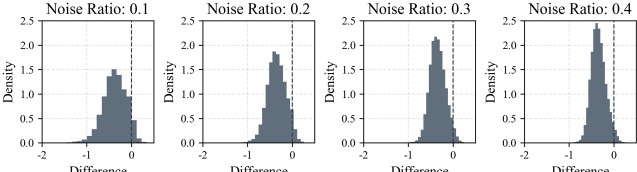

Figure 4: Difference between normal and noisy interactions in personal loss distributions across all users.

significant overlap in the overall loss distribution across varying noise ratios. As shown in Table 1, across different noise ratios, the values of $|\mathcal{I}^{\mathcal{G}}_{\text{normal}}|/|\mathcal{I}_{\text{normal}}|$ and $|\mathcal{I}^{\mathcal{G}}_{\text{noise}}|/|\mathcal{I}_{\text{noise}}|$ are generally high. This makes it difficult to distinguish between normal and noisy interactions based on the overall loss distribution, increasing the likelihood of denoising errors in existing methods [5, 8, 15, 26]. Consequently, relying solely on the overall loss distribution may not be an effective approach for differentiating between normal and noisy interactions.

**User Case in Personal Loss Distribution.** To further analyze these interactions, we randomly select five users from the dataset and display their personal loss distributions across varying noise ratios. As shown in Figure 3, for each user, the losses of normal interactions consistently remain lower than those of noisy interactions. However, due to significant variance in users' personal loss distributions, the overlap depicted in Figure 2 is primarily attributed to certain users exhibiting normal interaction losses that exceed other users' noisy interaction losses. For example, at a noise ratio of 0.2, the noisy interaction losses of user 2 differ from corresponding normal interaction losses but are similar to the normal interaction losses of user 5.

**Statistics of Personal Loss Distribution.** Building on the previous user case analysis, we propose that noisy interactions can be more effectively identified by analyzing users' personal loss distributions. To further illustrate this, we examine the statistical differences between users' normal and noisy interaction losses. For each user, we compute the difference between the lower quartile of their normal interaction losses and the upper quartile of their noisy interaction losses. As depicted in Figure 4, most users exhibit higher noisy interaction losses compared to their normal interaction losses, a trend that persists across all noise levels. As shown in Table 2, compared to $|\mathcal{I}^{\mathcal{G}}_{\text{normal}}|$ and $|\mathcal{I}^{\mathcal{G}}_{\text{noise}}|$, $|\mathcal{I}^{\mathcal{P}}_{\text{normal}}|$ and $|\mathcal{I}^{\mathcal{P}}_{\text{noise}}|$ decrease significantly, further validating the effectiveness of personal loss distributions for distinguishing normal interactions from noisy ones. This observation offers valuable insights for potential improvements in denoising strategies.

## 4.2 PLD Methodology

Based on the above insights, a straightforward denoising method would treat higher-loss interactions within the personal loss distribution as noise. However, the sparsity of user interactions causes significant fluctuations in personal loss distributions. As a result, reweight-based methods may cause drastic changes in the weight assigned to the same interaction across consecutive epochs, undermining training stability. Additionally, due to variations in the presence and amount of noise, dropping the highest-loss interactions could negatively affect users with little or no noisy interactions. For

Table 2: Statistics of personal loss distribution

| Noise Ratio | $|\mathcal{I}^{\mathcal{P}}_{\text{normal}}|$ | $|\mathcal{I}^{\mathcal{P}}_{\text{normal}}|/|\mathcal{I}_{\text{normal}}|$ | $|\mathcal{I}^{\mathcal{P}}_{\text{noise}}|$ | $|\mathcal{I}^{\mathcal{P}}_{\text{noise}}|/|\mathcal{I}_{\text{noise}}|$ |
|---|---|---|---|---|
| 0.1 | 38,998 | 5.13% | 2,125 | 2.79% |
| 0.2 | 36,971 | 4.44% | 4,979 | 2.99% |
| 0.3 | 35,571 | 4.28% | 7,969 | 3.19% |
| 0.4 | 35,511 | 4.27% | 10,771 | 3.24% |

instance, with a fixed drop rate (e.g., 10%), a user without noisy interactions would still experience a 10% drop in normal interactions during training, which would degrade the user's experience.

To address these issues, we propose solving this problem through probabilistic sampling. Specifically, we aim to reduce the probability of noisy interactions being optimized while ensuring that users without noise remain unaffected. To this end, we propose a resampling strategy named PLD, which consists of two parts: Candidate Pool Construction and Item Resampling.

**Candidate Pool Construction.** To prevent items with extremely small losses from being repeatedly sampled, we pre-construct a candidate item pool, $C_u^k$ of size $k$ for each user $u$. Items in $C_u^k$ are randomly sampled from the user's interacted items, $\mathcal{V}_u$.

**Item Resampling.** Next, we calculate the loss $l_{u,v}$ for each of the $k$ items in the candidate pool. We then perform resampling based on the computed loss values. Specifically, for user $u$, the sampling probability for item $v$ in the candidate pool $C_u^k$ is determined by:

$$P_{u,v} = \frac{\exp(-l_{u,v})}{\sum_{j \in C_u^k} \exp(-l_{u,j})}. \tag{1}$$

Finally, the resampled item is selected as the positive interaction for the current optimization step.

This method ensures that variances in personal loss distributions do not adversely affect the sampling process. Moreover, this approach ensures that normal interactions are optimized, even for users without noisy interactions—unlike previous methods, which always drop a subset of interactions [8, 26].

## 4.3 Theoretical Analysis

To analyze the effectiveness of the PLD method, we examine the probability that PLD samples both normal and noisy interactions.

THEOREM 1. *For a user $u$, there are $n$ items with normal interactions and $m$ items with noisy interactions. Assume that the loss of the user's normal interactions follows a Gaussian distribution $\mathcal{N}(\mu_1, \sigma^2)$ and the loss of noisy interactions follows a Gaussian distribution $\mathcal{N}(\mu_2, \sigma^2)$, where $\mu_1 < \mu_2$ and $\mu_1, \mu_2 > \sigma$. From these $m + n$ interactions, we first randomly select $k$ interactions, and then resample one positive interaction according to Equation 1. Let $\Lambda_{normal}$ denote the sum of sampling probabilities for normal interactions, and $\Lambda_{noise}$ denote the sum of sampling probabilities for noisy interactions. Define*

**Algorithm 1** Training Procedure with PLD

---

**Input:** Training set $\mathcal{D}$, pool size $k$, temperature coefficient $\tau$, batch size $\mathbb{B}$, loss function $\mathcal{L}(u, i, j)$

**Output:** Model parameters $\Theta$.

1: **while** stopping criteria not met **do**
   ▷ *PLD*
2:   Draw $\mathbb{B}$ triples $(u, C_u^k, j)$ from $\mathcal{D}$.
3:   Initialize the batch set $\mathcal{D}_{\mathbb{B}} = \emptyset$
4:   **for** each $(u, C_u^k, j)$ **do**
5:     Calculate $l_i$ for $i \in C_u^k$ using $\mathcal{L}(u, i, j)$.
6:     Resample $i^*$ based on Equation 3 within $C_u^k$.
7:     Add $(u, i^*, j)$ to the batch set $\mathcal{D}_{\mathbb{B}}$.
8:   **end for**
   ▷ *Standard Training*
9:   Update $\Theta$ according to $\mathcal{L}(u, i, j)$ for each $(u, i^*, j)$ in $\mathcal{D}_{\mathbb{B}}$.
10: **end while**
11: **return** $\Theta$

---

*the following:*

$$\alpha = \exp\left(-\mu_1 + \frac{\sigma^2}{2}\right), \quad \beta = \exp\left(-\mu_2 + \frac{\sigma^2}{2}\right),$$

$$\gamma = \exp(\sigma^2) - 1, \qquad\qquad \eta = \frac{n\alpha + m\beta}{n + m},$$

$$\Gamma = \frac{(n\alpha - m\beta)}{m + n} \cdot \frac{(\alpha^2 + \beta^2)(\gamma + \frac{m}{n+m}) + \beta^2}{\eta^3},$$

$$\chi = \frac{\gamma}{(n + m)}\left[n\alpha^2 - m\beta^2\right]$$

*we have:*

$$\mathbb{E}[\Lambda_{normal} - \Lambda_{noise}] = \begin{cases} \dfrac{n - m}{n + m} & k = 1 \\[2mm] \dfrac{n\alpha - m\beta}{(m+n)\eta} + \underbrace{\dfrac{\Gamma}{k} - \dfrac{\chi}{C^2}\dfrac{k}{(k-1)^2}}_{\text{Fluctuation term}} & k > 1 \end{cases},$$
(2)

*where $C \in [\beta, \alpha]$ is a constant term.*

The proof of Theorem 1 is detailed in Appendix A.1. The term $\frac{\Gamma}{k} - \frac{\chi}{C^2}\frac{k}{(k-1)^2}$ arises from the variance component in the denominator of the softmax function, exhibiting larger fluctuations when $k$ is small, while stabilizing as $k$ increases.

According to Theorem 1, when $k = 1$, PLD reduces to standard training with $\mathbb{E}[\Lambda_{normal} - \Lambda_{noise}] = \frac{n-m}{n+m}$. For $k > 1$, given $\alpha, \beta, \gamma > 0$, with $\alpha > \beta$ and $n \gg m$, we find that $\Gamma > \frac{\chi}{C^2}$. Thus, $\mathbb{E}[\Lambda_{normal} - \Lambda_{noise}] > \frac{n-m}{n+m}$. This indicates that **PLD outperforms standard training, demonstrating superior denoising capabilities.**

To further enhance the effectiveness of the PLD method, we can increase $\frac{n\alpha - m\beta}{(m+n)\eta}$. Specifically, let $\xi = \frac{\beta}{\alpha} = \exp(\mu_1 - \mu_2) < 1$. Then, we can express $\frac{n\alpha - m\beta}{(m+n)\eta} = \frac{n - \xi m}{n + \xi m}$. Notably, since $\frac{\partial \frac{n\alpha - m\beta}{(m+n)\eta}}{\partial \xi} < 0$, we can decrease $\xi$ to amplify $\frac{n\alpha - m\beta}{(m+n)\eta}$, thus enlarging $\mathbb{E}[\Lambda_{normal} - \Lambda_{noise}]$. Based on this idea, we introduce a temperature coefficient $\tau$ into Equation 1:

$$P_{u,v} = \frac{\exp(-l_{u,v}/\tau)}{\sum_{j \in C_u^k} \exp(-l_{u,j}/\tau)}.$$
(3)

**Table 3: Dataset statistics**

| DATASET | #Users | #Items | #Interactions | Avg.Inter. | Sparsity |
|---|---|---|---|---|---|
| Gowalla | 29,858 | 40,981 | 1,027,370 | 34.4 | 99.92% |
| Yelp2018 | 31,668 | 38,048 | 1,561,406 | 49.3 | 99.88% |
| MIND | 38,441 | 38,000 | 1,210,953 | 31.5 | 99.92% |
| MIND-Large | 111,664 | 54,367 | 3,294,424 | 29.5 | 99.95% |

In this manner, the new $\xi'$ can be considered as $\xi' = \exp((\mu_1 - \mu_2)/\tau)$. By reducing $\tau$, we can further enlarge $\frac{n\alpha - m\beta}{(m+n)\eta}$. The algorithmic flow of PLD is outlined in Algorithm 1.

Additionally, we perform an in-depth analysis and comparison of the time and space complexity of PLD and baseline methods. For further details, please refer to Appendix A.2.

## 5 EXPERIMENTS

In this section, we conduct extensive experiments to address the following research questions (**RQs**):

- **RQ1:** How does PLD perform compared to state-of-the-art denoising methods?
- **RQ2:** How well does PLD generalize, align with the theoretical analysis, and what is its time complexity?
- **RQ3:** How do the hyperparameters affect the performance of PLD?

### 5.1 Experimental Setup

*5.1.1 Datasets.* We utilize four widely recognized datasets: the **Gowalla** check-in dataset [14], the **Yelp2018** business dataset, and the **MIND** and **MIND-Large** news recommendation datasets [31]. The Gowalla and Yelp2018 datasets include all users, while for the MIND dataset, we sample two subsets of users, constructing MIND and MIND-Large, following [42]. Consistent with [27, 41], we exclude users and items with fewer than 10 interactions from our analysis. We allocate 80% of each user's historical interactions to the training set, reserving the remainder for testing. Additionally, 10% of the training set is randomly selected to form a validation set for hyperparameter tuning. Detailed statistics for the datasets are summarized in Table 3.

*5.1.2 Baselines.* We incorporate various denoising methods, including four reweight-based approaches and one self-supervised method. Specifically, we evaluate R-CE, T-CE [26], BOD [30], and DCF [8] as reweight-based methods, and DeCA [29] as a self-supervised method.

- **R-CE** [26]: R-CE assigns reduced training weight to high-loss interactions.
- **T-CE** [26]: T-CE drops interactions with the highest loss values at a predefined drop rate.
- **BOD** [30]: BOD treats the process of determining interaction weights as a bi-level optimization problem to learn more effective denoising weights.
- **DCF** [8]: DCF addresses the challenges posed by hard positive samples and the data sparsity introduced by dropping interactions in T-CE.
- **DeCA** [29]: DeCA posits that clean samples tend to yield consistent predictions across different models, incorporating two recommendation models during training to better differentiate between clean and noisy interactions.

**Table 4: Recommendation performance of different denoising methods. The highest scores are in bold, and the runner-ups are with underlines. A significant improvement over the runner-up is marked with * (i.e., two-sided t-test with $0.05 \leq p < 0.1$) and ** (i.e., two-sided t-test with $p < 0.05$).**

| Model | Gowalla | | | | Yelp2018 | | | | MIND | | | |
|---|---|---|---|---|---|---|---|---|---|---|---|---|
| | Recall | | NDCG | | Recall | | NDCG | | Recall | | NDCG | |
| | @20 | @50 | @20 | @50 | @20 | @50 | @20 | @50 | @20 | @50 | @20 | @50 |
| **MF** | 0.1486 | 0.2410 | 0.1073 | 0.1370 | 0.0621 | 0.1187 | 0.0483 | 0.0704 | 0.0658 | 0.1219 | 0.0430 | 0.0615 |
| +R-CE | 0.1456 | 0.2362 | 0.1053 | 0.1343 | 0.0654 | 0.1239 | 0.0506 | 0.0733 | 0.0716 | 0.1311 | 0.0468 | 0.0663 |
| +T-CE | 0.1326 | 0.2197 | 0.0920 | 0.1197 | 0.0571 | 0.1113 | 0.0430 | 0.0639 | 0.0359 | 0.0812 | 0.0215 | 0.0363 |
| +DeCA | 0.1463 | 0.2356 | 0.1068 | 0.1355 | 0.0645 | 0.1225 | 0.0502 | 0.0729 | 0.0714 | 0.1312 | 0.0471 | 0.0668 |
| +BOD | 0.1489 | 0.2415 | 0.1079 | 0.1376 | 0.0654 | 0.1235 | 0.0511 | 0.0738 | 0.0713 | 0.1300 | 0.0473 | 0.0665 |
| +DCF | 0.1489 | 0.2413 | 0.1073 | 0.1367 | 0.0635 | 0.1208 | 0.0493 | 0.0715 | 0.0710 | 0.1297 | 0.0472 | 0.0665 |
| +PLD (ours) | **0.1520**\*\* | **0.2475**\*\* | **0.1097** | **0.1404**\* | **0.0677**\*\* | **0.1264**\*\* | **0.0527**\*\* | **0.0755**\*\* | **0.0769**\*\* | **0.1379**\*\* | **0.0513**\* | **0.0713**\*\* |
| Gain | +2.04% ↑ | +2.47% ↑ | +1.71% ↑ | +1.98% ↑ | +3.46% ↑ | +2.02% ↑ | +3.09% ↑ | +2.33% ↑ | +7.36% ↑ | +5.09% ↑ | +8.46% ↑ | +6.83% ↑ |
| **LightGCN** | 0.1553 | 0.2509 | 0.1142 | 0.1449 | 0.0665 | 0.1270 | 0.0516 | 0.0750 | 0.0817 | 0.1485 | 0.0538 | 0.0757 |
| +R-CE | 0.1536 | 0.2481 | 0.1131 | 0.1434 | 0.0554 | 0.1042 | 0.0428 | 0.0617 | 0.0723 | 0.1315 | 0.0478 | 0.0670 |
| +T-CE | 0.1146 | 0.1859 | 0.0859 | 0.1088 | 0.0532 | 0.1004 | 0.0412 | 0.0595 | 0.0674 | 0.1222 | 0.0447 | 0.0626 |
| +DeCA | 0.1540 | 0.2495 | 0.1133 | 0.1440 | 0.0678 | 0.1298 | 0.0526 | 0.0766 | 0.0812 | 0.1480 | 0.0532 | 0.0751 |
| +BOD | 0.1560 | 0.2519 | 0.1154 | 0.1461 | 0.0672 | 0.1280 | 0.0523 | 0.0758 | 0.0809 | 0.1475 | 0.0532 | 0.0750 |
| +DCF | 0.1276 | 0.2072 | 0.0948 | 0.1203 | 0.0619 | 0.1180 | 0.0482 | 0.0699 | 0.0734 | 0.1342 | 0.0483 | 0.0681 |
| +PLD (ours) | **0.1580**\*\* | **0.2558**\*\* | **0.1157** | **0.1472**\* | **0.0693**\*\* | **0.1325**\*\* | **0.0538**\*\* | **0.0783**\*\* | **0.0837**\*\* | **0.1516**\*\* | **0.0551**\*\* | **0.0774**\*\* |
| Gain | +1.23% ↑ | +1.53% ↑ | +0.32% ↑ | +0.71% ↑ | +2.31% ↑ | +2.02% ↑ | +2.44% ↑ | +2.22% ↑ | +2.43% ↑ | +2.06% ↑ | +2.47% ↑ | +2.33% ↑ |

**Table 5: Recommendation performance of different denoising methods on MIND-Large.**

| Model | Recall | | NDCG | |
|---|---|---|---|---|
| | @20 | @50 | @20 | @50 |
| **MF** | 0.0788 | 0.1441 | 0.0501 | 0.0710 |
| +R-CE | 0.0790 | 0.1453 | 0.0502 | 0.0715 |
| +T-CE | 0.0329 | 0.0788 | 0.0194 | 0.0340 |
| +DeCA | 0.0793 | 0.1447 | 0.0507 | 0.0717 |
| +BOD | 0.0794 | 0.1435 | 0.0516 | 0.0722 |
| +DCF | 0.0818 | 0.1482 | 0.0528 | 0.0741 |
| +PLD (ours) | **0.0846**\*\* | **0.1524**\*\* | **0.0543**\*\* | **0.0760**\*\* |
| Gain | +3.36% ↑ | +2.85% ↑ | +2.67% ↑ | +2.58% ↑ |

**Table 6: Recommendation performance of different denoising methods on BCE loss on MIND.**

| Model | Recall | | NDCG | |
|---|---|---|---|---|
| | @20 | @50 | @20 | @50 |
| **MF** | 0.0585 | 0.1156 | 0.0337 | 0.0520 |
| +R-CE | 0.0644 | 0.1235 | 0.0387 | 0.0581 |
| +T-CE | 0.0612 | 0.1193 | 0.0361 | 0.0550 |
| +DeCA | 0.0655 | 0.1250 | 0.0394 | 0.0588 |
| +BOD | 0.0681 | 0.1243 | 0.0442 | 0.0624 |
| +DCF | 0.0683 | 0.1319 | 0.0407 | 0.0615 |
| +PLD (ours) | **0.0731**\*\* | **0.1334** | **0.0464**\*\* | **0.0663**\*\* |
| Gain | +6.98% ↑ | +1.16% ↑ | +4.94% ↑ | +6.13% ↑ |

*5.1.3 Evaluation Metrics.* We adopt standard metrics widely employed in the field. The primary metrics for evaluating recommendation performance are the top-$k$ metrics: Recall at $K$ (Recall@$K$) and Normalized Discounted Cumulative Gain at $K$ (NDCG@$K$), as described in [6, 9, 40]. For evaluation, we set $K = 20$ and $K = 50$, following [27, 41].

*5.1.4 Implementation Details.* In our study, we employ two commonly used backbone recommendation models: MF [11] and Light-GCN [6]. The configuration of both denoising methods and recommendation models involves selecting a learning rate from {0.1, 0.01, ..., $1 \times 10^{-5}$}, and a weight decay from {0, 0.1, ..., $1 \times 10^{-5}$}. For PLD, the candidate pool size $k$ is selected from {2, 3, 5, 10, 20}, and the temperature coefficient $\tau$ is chosen from {0.01, 0.05, 0.1, 0.2, 0.3, 0.4, 0.5}. For the baselines, hyperparameter settings follow those specified in the original publications. Our implementation code is available at the following link[2].

## 5.2 Performance Comparison (RQ1)

In this section, we address **RQ1** by focusing on two key aspects: recommendation performance and robustness against noise. All

[2]https://anonymous.4open.science/r/PLD

results in this section are based on the widely adopted BPR loss function [18]. For a comprehensive evaluation, results using the BCE loss function are provided in Section 5.3.

**Recommendation Performance.** We evaluate the effectiveness of PLD across three common datasets without introducing additional noise, as shown in Table 4. The performance of R-CE [26], T-CE [26], and DCF [8] is suboptimal due to the limitations of using overall loss distribution as a denoising criterion for pairwise loss functions, as discussed earlier (in Figure 1). In particular, T-CE applies a fixed drop ratio, which truncates part of the loss completely, unintentionally discarding many normal interactions and leading to a significant performance decrease.

On the other hand, DeCA [29] and BOD [30] demonstrate more stable performance, securing runner-up results across several metrics. **Our method**, PLD, mitigates the impact of noisy interactions by resampling based on users' personal loss distributions, producing stable and optimal results across all datasets. It achieves significant improvements, with 4.29% and 4.42% increase in Recall@20 and NDCG@20, respectively, using MF as the backbone model.

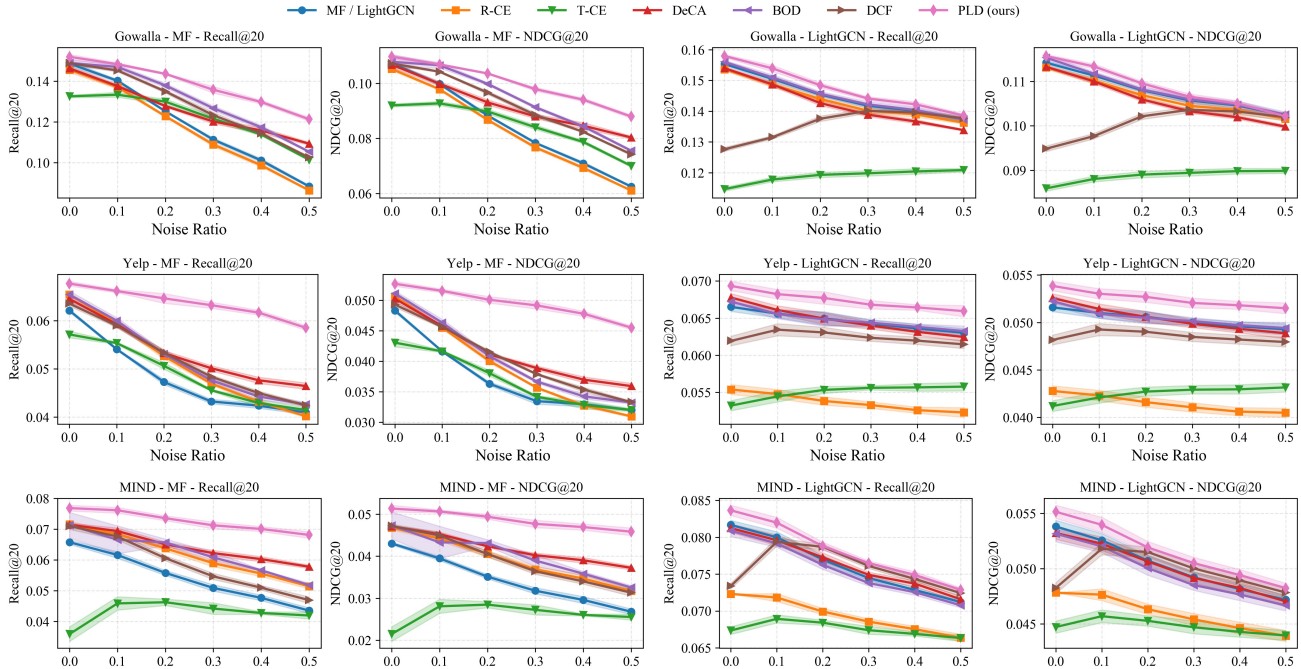

Figure 5: Recommendation performance of different denoising methods across various noise ratios.

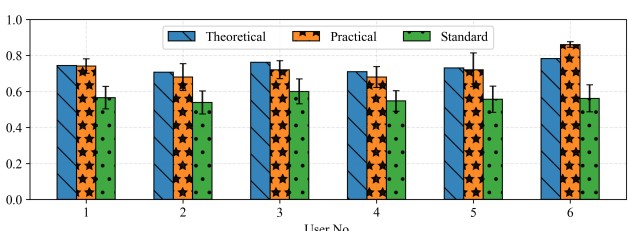

Figure 6: Theoretical, practical, and standard values (i.e., without resampling in PLD) of $\mathbb{E}[\Lambda_{normal} - \Lambda_{noise}]$ for 6 users on MIND with 30% additional noise.

**Robustness against Noise.** We further assess PLD's robustness to noise by randomly introducing noisy interactions at ratios[3] ranging from 0.1 to 0.5, as shown in Figure 5. As the noise ratio increases, the performance of all methods decreases. Additionally, we observe that in some cases, specifically when using LightGCN as the backbone model, denoising methods based solely on overall loss distribution (T-CE, R-CE, and DCF) perform worse than the backbone model itself. This further confirms that overall loss distribution is unsuitable for denoising in pairwise loss scenarios.

In contrast, our method, PLD, remains the most stable across all noise ratios, consistently outperforming other denoising methods. Additionally, we show the results on a larger dataset, MIND-Large (Table 5), where only the results at a noise ratio of 0.1 are presented due to space limitations. The conclusions drawn from MIND-Large are consistent with those from the other datasets.

---

[3]A ratio of 0.1 means adding noisy interactions equal to $10\%|\mathcal{I}_{normal}|$.

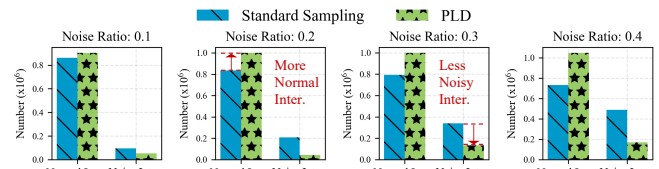

Figure 7: Number of normal interactions and noisy interactions sampled.

## 5.3 Argumentation Study (RQ2)

In this section, we address **RQ2** by evaluating the generalization of our method with the pointwise loss function, analyzing the consistency between Theorem 1 and practical results, and verifying the advantage of our method in terms of time complexity.

**Pointwise Loss Function.** To further demonstrate the generalization of PLD, we evaluate its performance using the pointwise loss function, specifically Binary Cross-Entropy (BCE) loss. Table 6 presents the results with MF as the backbone model on the MIND dataset. Unlike the results with pairwise loss functions shown in Table 4, all denoising methods show improvements under MF with the BCE loss function, particularly T-CE. Since these methods are originally designed with BCE loss in mind, they perform well with BCE but struggle to adapt to BPR loss. In contrast, our method, PLD, not only adapts but also achieves the best results with BCE loss, showing a 6.98% improvement in Recall@20 and a 4.94% improvement in NDCG@20.

**Theorem Validation.** To evaluate the consistency between Theorem 1 and practical results, and thereby demonstrate the effectiveness of PLD, we examine the alignment between the theoretical value $\mathbb{E}[\Lambda_{normal} - \Lambda_{noise}]$ and its practical counterpart. We also

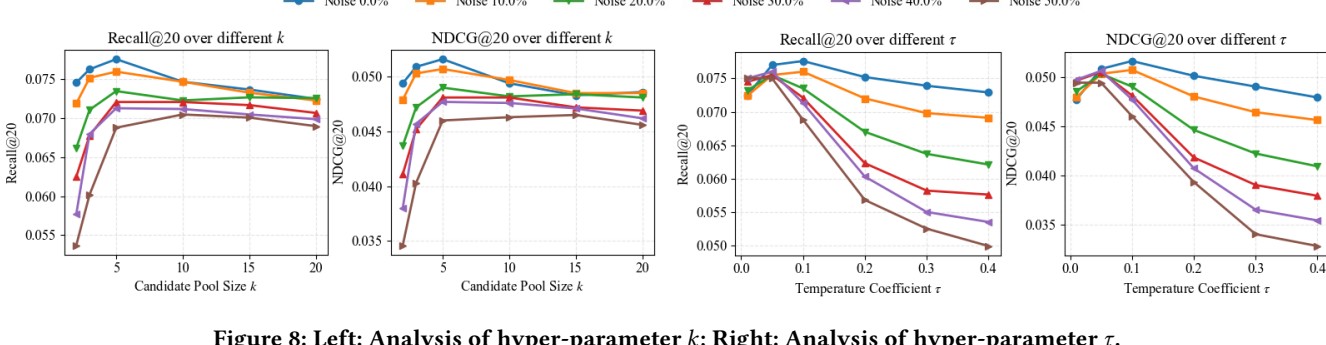



Figure 8: Left: Analysis of hyper-parameter $k$; Right: Analysis of hyper-parameter $\tau$.

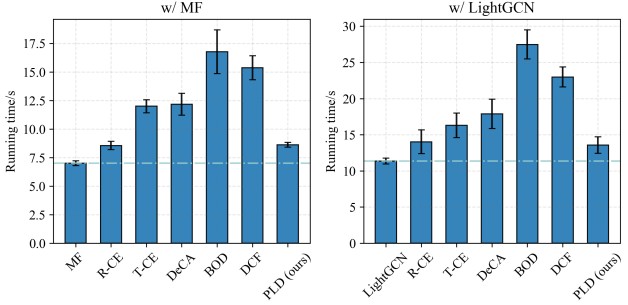

**Figure 9: Training time per epoch (in seconds) with batch size 2048 on MIND.**

compare these values to the probability values under a standard training process without resampling. Since Theorem 1 contains a constant $C \in [\alpha, \beta]$, we approximate it by setting $C = \frac{\alpha+\beta}{2}$ for probability calculations.

We randomly selecte 6 users from the dataset and use Equation 2 to calculate $\mathbb{E}[\Lambda_{\text{normal}} - \Lambda_{\text{noise}}]$. Concurrently, we compute the practical value through 100 simulations of the sampling process in lines 5-7 of Algorithm 1. Finally, we obtain the standard value by running 100 simulations using the standard training process without resampling.

As shown in Figure 6, the theoretical value of $\mathbb{E}[\Lambda_{\text{normal}} - \Lambda_{\text{noise}}]$ closely aligns with the practical value, verifying the correctness of our theoretical analysis. Moreover, we observe that the practical value corresponding to PLD is significantly higher than the standard value, highlighting the effectiveness of our method.

Additionally, we compare the number of normal and noisy interactions sampled in each epoch for PLD and standard training, as shown in Figure 7. PLD significantly reduces the number of noisy interactions sampled.

**Time Complexity.** To validate the advantage of our method in terms of time complexity, we compare the per-epoch runtime of baseline methods on the MIND dataset. Training is conducted on an RTX 4090, and we record the average training time over 100 epochs. To avoid GPU memory limitations, we standardize the batch size to 2048, which reduces the number of sorting operations within each batch, thereby lowering the time complexity for methods like T-CE [26] and DCF [8] that rely on batch-level sorting.

As shown in Figure 9, BOD [30] incurs additional time costs due to the extra training required for the weight encoder and decoder.

T-CE and DCF require sorting the loss within each batch, leading to higher time costs. DeCA [29] involves training multiple models, further increasing time overhead. In contrast, both R-CE [26] and **our method**, PLD, do not significantly increase time complexity, as their time is close to that of the backbone model.

## 5.4 Hyper-Parameters Analysis (RQ3)

In this section, we address **RQ3** by exploring the effects of hyperparameters on MIND with MF as the backbone model, specifically the candidate pool size $k$ and the temperature coefficient $\tau$. The results are shown in Figure 8.

**Analysis of Hyper-Parameter $k$.** With $\tau$ fixed at 0.1, we vary $k$ within the range $[2, 3, 5, 10, 15, 20]$. We observe that when the candidate pool size is too small, i.e., $k = 2$, the high sampling variance often results in the candidate pool being dominated by noisy samples. At $k = 5$, the method consistently achieves good performance across all noise ratios. Beyond $k = 10$, the performance stabilizes, showing minimal additional improvements.

**Analysis of Hyper-Parameter $\tau$.** With $k$ set to 5, we vary $\tau$ within the range $[0.01, 0.05, 0.1, 0.2, 0.3, 0.4]$. We find that when $\tau$ is large, i.e., $\tau \geq 0.2$, the performance of PLD fluctuates significantly. In contrast, when $\tau \leq 0.1$, the performance becomes more stable. Specifically, at $\tau = 0.05$, the results exhibit smaller variations across different noise ratios, indicating that PLD has a stronger denoising effect under this configuration.

## 6 CONCLUSION

In this research, we identify the limitations of denoising indicators used in current loss-based denoising methods, particularly the significant overlap between normal and noisy interactions in the overall loss distribution. Our analysis reveals a clear distinction between normal and noisy interactions in users' personal loss distributions. Building on these findings, we introduce a novel denoising strategy, PLD, which incorporates a resampling approach based on users' personal loss distributions. By selectively resampling training interactions, PLD effectively reduces the likelihood of noisy interactions being optimized. Additionally, we conduct a comprehensive theoretical analysis, demonstrating the robustness of PLD and suggesting potential ways to further enhance its performance. Extensive experimental results confirm the strong efficacy and robustness of PLD in denoising recommender systems.

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

# A APPENDIX

## A.1 Proofs

PROPOSITION 1. *Let* $x_i \sim \mathcal{N}(\mu, \sigma^2)$ *and* $N \sim \text{Binomial}(k, \frac{n}{n+m})$. *Define*

$$S_x = \sum_{i=1}^{N} \exp(-x_i).$$

*Then, the expected value and variance of* $S_x$ *are given by:*

$$\mathbb{E}[S_x] = \frac{kn}{m+n} \exp\left(-\mu + \frac{\sigma^2}{2}\right),$$

$$\text{Var}[S_x] = \frac{kn}{n+m} \exp(-2\mu + \sigma^2)\left(\exp(\sigma^2) - \frac{n}{n+m}\right).$$

PROOF. To compute $\mathbb{E}[S_x]$, we apply the Double Expectation Theorem [19]. First, we condition on $N$:

$$\mathbb{E}[S_x] = \mathbb{E}_N\left[\mathbb{E}_{S_x}[S_x \mid N]\right]$$

$$= \mathbb{E}_N\left[N \exp\left(-\mu + \frac{\sigma^2}{2}\right)\right].$$

The inner expectation evaluates to $N \exp\left(-\mu + \frac{\sigma^2}{2}\right)$ since $\mathbb{E}[\exp(-x_i)]$ for each $x_i \sim \mathcal{N}(\mu, \sigma^2)$ is known. Taking the expectation over $N \sim \text{Binomial}(k, \frac{n}{n+m})$, we obtain:

$$\mathbb{E}[S_x] = \frac{kn}{m+n} \exp\left(-\mu + \frac{\sigma^2}{2}\right).$$

Next, we compute the variance of $S_x$ using the Law of Total Variance [1]:

$$\text{Var}[S_x] = \mathbb{E}_N\left[\text{Var}_{S_x}[S_x \mid N]\right] + \text{Var}_N\left[\mathbb{E}_{S_x}[S_x \mid N]\right].$$

For the first term, $\text{Var}_{S_x}[S_x \mid N] = N \exp(-2\mu + \sigma^2)\left(\exp(\sigma^2) - 1\right)$, leading to:

$$\mathbb{E}_N\left[\text{Var}_{S_x}[S_x \mid N]\right] = \mathbb{E}[N] \exp(-2\mu + \sigma^2)\left(\exp(\sigma^2) - 1\right).$$

For the second term, we use the variance of $N$, yielding:

$$\text{Var}_N\left[\mathbb{E}_{S_x}[S_x \mid N]\right] = \text{Var}[N] \exp(-2\mu + \sigma^2).$$

Substituting $\mathbb{E}[N] = \frac{kn}{n+m}$ and $\text{Var}[N] = \frac{knm}{(n+m)^2}$, we obtain the final expression:

$$\text{Var}[S_x] = \frac{kn}{n+m} \exp(-2\mu + \sigma^2)\left(\exp(\sigma^2) - \frac{n}{n+m}\right).$$

$\square$

PROPOSITION 2. *Given two independent random variables* $X$ *and* $Y$, *we have*

$$\mathbb{E}\left[\frac{1}{X+Y}\right] \approx \frac{1}{\mathbb{E}[X] + \mathbb{E}[Y]}\left(1 + \frac{\text{Var}[X] + \text{Var}[Y]}{(\mathbb{E}[X] + \mathbb{E}[Y])^2}\right).$$

PROOF. Let $Z = X+Y$ and define $g(Z) = \frac{1}{Z}$. Applying the second-order Taylor expansion [13] of $g(Z)$ around $\mathbb{E}[Z]$, we obtain:

$$g(Z) \approx g(\mathbb{E}[Z]) + g'(\mathbb{E}[Z])(Z - \mathbb{E}[Z]) + \frac{1}{2}g''(\mathbb{E}[Z])(Z - \mathbb{E}[Z])^2.$$

Taking the expectation of both sides, the linear term vanishes due to $\mathbb{E}[Z - \mathbb{E}[Z]] = 0$, leaving:

$$\mathbb{E}[g(Z)] \approx g(\mathbb{E}[Z]) + \frac{1}{2}g''(\mathbb{E}[Z])\mathbb{E}\left[(Z - \mathbb{E}[Z])^2\right]$$

$$= g(\mathbb{E}[Z]) + \frac{1}{2}g''(\mathbb{E}[Z])\text{Var}[Z].$$

Substituting $g(Z) = \frac{1}{Z}$, we have $g'(\mathbb{E}[Z]) = -\frac{1}{\mathbb{E}[Z]^2}$ and $g''(\mathbb{E}[Z]) = \frac{2}{\mathbb{E}[Z]^3}$. Thus, the expression simplifies to:

$$\mathbb{E}[g(Z)] \approx \frac{1}{\mathbb{E}[Z]} + \frac{1}{2}\frac{2}{\mathbb{E}[Z]^3}\text{Var}[Z]$$

$$= \frac{1}{\mathbb{E}[Z]}\left(1 + \frac{\text{Var}[Z]}{\mathbb{E}[Z]^2}\right).$$

Since $Z = X+Y$ and $X$ and $Y$ are independent, we use the properties $\mathbb{E}[Z] = \mathbb{E}[X] + \mathbb{E}[Y]$ and $\text{Var}[Z] = \text{Var}[X] + \text{Var}[Y]$. Substituting these into the above expression gives:

$$\mathbb{E}\left[\frac{1}{X+Y}\right] \approx \frac{1}{\mathbb{E}[X] + \mathbb{E}[Y]}\left(1 + \frac{\text{Var}[X] + \text{Var}[Y]}{(\mathbb{E}[X] + \mathbb{E}[Y])^2}\right),$$

which completes the proof. $\square$

PROOF OF THEOREM 1. For $k$ samples, let $N$ be the number of normal samples and $M$ be the number of noisy samples. We have:

$$\mathbb{E}[N] = k \cdot \frac{n}{n+m}, \qquad \text{Var}[N] = k \cdot \frac{nm}{(n+m)^2},$$

$$\mathbb{E}[M] = k \cdot \frac{m}{n+m}, \qquad \text{Var}[M] = k \cdot \frac{nm}{(n+m)^2}.$$

When $k = 1$, the sampling probabilities simplify to:

$$P_i = P_j = \frac{1}{n+m}.$$

Therefore, the expected difference in sampling probabilities is:

$$\mathbb{E}[\Lambda_{\text{normal}} - \Lambda_{\text{noise}}] = \frac{n-m}{n+m}.$$

Now, for $k > 1$, let $x_i \sim \mathcal{N}(\mu_1, \sigma_1^2)$ represent the loss of a normal sample $i$, and $y_j \sim \mathcal{N}(\mu_2, \sigma^2)$ represent the loss of a noisy sample $j$. According to Equation 1, the probability of selecting sample $i$ is:

$$P_i = \frac{\exp(-x_i)}{\sum_{i=1}^{N} \exp(-x_i) + \sum_{j=1}^{M} \exp(-y_j)}.$$

Define:

$$S_x = \sum_{i=1}^{N} \exp(-x_i), \quad S_y = \sum_{j=1}^{M} \exp(-y_j).$$

The sum of the sampling probabilities of normal interactions becomes:

$$\Lambda_{\text{normal}} = \sum_{i=1}^{N} P_i = \frac{S_x}{S_x + S_y}.$$

Then, we have:

$$\mathbb{E}[\Lambda_{\text{normal}}] = \mathbb{E}[S_x] \cdot \mathbb{E}\left[\frac{1}{S_x + S_y}\right] + \text{Cov}\left(S_x, \frac{1}{S_x + S_y}\right).$$

**Table 7: Method complexity comparison.**

| Methods | Space Complexity | Time Complexity |
|---------|------------------|-----------------|
| Base | $M$ | $O(N)$ |
| T-CE | $M$ | $O(N \log(N))$ |
| BOD | $M + d_1 \times d_2$ | $O((d_1 \times d_2)N)$ |
| DCF | $M$ | $O(N \log(N))$ |
| **PLD (ours)** | $M$ | $O(kN)$ |

Expanding the covariance term:

$$\text{Cov}\left(S_x, \frac{1}{S_x + S_y}\right) = \mathbb{E}_N\left[\text{Cov}\left(\sum_{i=1}^{N} \exp(-x_i) \mid N, \frac{1}{S_x + S_y}\right)\right]$$
$$+ \text{Cov}\left(\mathbb{E}\left[\sum_{i=1}^{N} \exp(-x_i) \mid N\right], \mathbb{E}\left[\frac{1}{S_x + S_y}\right]\right). \tag{4}$$

Assuming a linear dependence between $S_x + S_y$ and $\exp(-x_i)$, we introduce a constant $C \in [\beta, \alpha]$ such that:

$$S_x + S_y \approx \exp(-x_i) + (k-1)C.$$

This leads to:

$$\text{Cov}\left(\exp(-x_i), \frac{1}{S_x + S_y}\right) \approx \text{Cov}\left(\exp(-x_i), \frac{1}{\exp(-x_i) + (k-1)C}\right).$$

Using the approximation:

$$\frac{1}{\exp(-x_i) + (k-1)C} \approx \frac{1}{(k-1)C} - \frac{\exp(-x_i)}{(k-1)^2 C^2},$$

we find:

$$\text{Cov}\left(\exp(-x_i), \frac{1}{S_x + S_y}\right) \approx \text{Cov}\left(\exp(-x_i), \frac{1}{(k-1)C}\right)$$
$$- \text{Cov}\left(\exp(-x_i), \frac{\exp(-x_i)}{(k-1)^2 C^2}\right)$$
$$= -\frac{\text{Var}\left[\exp(-x_i)\right]}{(k-1)^2 C^2}.$$

Consequently:

$$\mathbb{E}[\Lambda_{\text{normal}}] = \mathbb{E}[S_x] \cdot \mathbb{E}\left[\frac{1}{S_x + S_y}\right] - \frac{kn}{n+m} \cdot \frac{\text{Var}\left[\exp(-x_i)\right]}{(k-1)^2 C^2}.$$

Finally, applying the definitions of $\alpha, \beta, \gamma, \eta, \Gamma,$ and $\chi$, we derive the expression for $\mathbb{E}[\Lambda_{\text{normal}} - \Lambda_{\text{noise}}]$ as:

$$\mathbb{E}[\Lambda_{\text{normal}} - \Lambda_{\text{noise}}] = \frac{n\alpha - m\beta}{(m+n)\eta} + \frac{\Gamma}{k} - \frac{\chi}{C^2}\frac{k}{(k-1)^2},$$

where the term $\frac{\chi}{C^2}\frac{k}{(k-1)^2}$ arises from the covariance component of the variance term. □

## A.2 Model Discussion

This section compares various reweight-based denoising methods, including T-CE [26], BOD [30], DCF [8], and our PLD, focusing on space and time complexities. The comparison is summarized in Table 7.

**Space Complexity.** The space complexity of the base model is determined by the number of parameters, denoted as $M$. T-CE, DCF, and our PLD do not introduce any additional modules, so their space complexity remains unchanged. In contrast, BOD introduces extra

components, specifically a generator and decoder (i.e., EN $\in \mathbb{R}^{d_1 \times d_2}$ and DE $\in \mathbb{R}^{d_2}$), which significantly increases its complexity.

**Time Complexity.** The time complexity of the base model is determined by the number of interactions, denoted as $N$, resulting in a complexity of $O(N)$. Both T-CE and DCF require sorting the loss values, increasing their complexity to $O(N \log N)$. BOD needs to encode and decode the weights of each edge, leading to a time complexity of $O((d_1 \times d_2 + d_1)N)$. Our PLD introduces a resampling process, adding an additional $O(2kN)$ to the time complexity, where $k \ll N$.

In summary, our PLD does not significantly increase the space or time complexity of the base model. Compared to other reweight-based denoising methods, our approach demonstrates clear advantages.

