# OpenReview forum: "Personalized Denoising Implicit Feedback for Robust Recommender System"
_ACM.org/TheWebConf/2025/Conference — WWW 2025 Poster_

### Official Review · Reviewer_is3p · 2024-11-29

**Novelty:** 6
**Technical Quality:** 5

**Review:**

This paper focuses on denoising user-item interaction behavior in recommender systems. Previous work often relied on the assumption that the atomic loss function of noisy interactions is higher than that of normal interactions, thereby identifying noisy interactions from raw interactions. However, this paper finds that the loss function value distribution of noisy interactions significantly overlaps with that of normal interactions. As a result, traditional methods tend to misclassify noisy interactions as normal, and vice versa. By analyzing the loss values of all interactions for each user, this paper proposes a personalized denoising strategy for interactions, called PLD. Through comprehensive experiments, ablation studies, and sound theoretical analysis, the proposed PLD significantly reduces the overlap between the loss function value distributions of noisy and normal interactions, while consistently improving recommendation performance.

### strengths:

* **S1:** This paper focuses on denoising user-item interaction behavior in recommender systems and proposes a new personalized strategy, PLD, to address the overlap between noisy and normal interaction loss distributions.
* **S2:** The proposed PLD strategy is based on analyzing the loss values of all interactions for each user, making it more tailored and effective for individual users.
* **S3:** Comprehensive experiments, thorough ablation studies, and solid theoretical analysis demonstrate that PLD significantly reduces the overlap between noisy and normal interaction loss distributions, and consistently improves recommendation performance by effectively handling noisy interactions, leading to more accurate and reliable recommendations.

**Questions:**

### weaknesses：

* **W1: potentially inappropriate assumption of homoscedastic Gaussian distribution in Theorem 1.** First of all, based on Figures 1 and 2, if we assume that the loss values follow a Gaussian distribution, the normal interaction and noisy interaction for overall losses are clearly not homoscedastic. Considering that the overall loss distribution is the expectation of the personalized loss distribution for each user, i.e., $l_{\text{overall}}(\text{interaction})=\mathbb{E}_{\text{user}}[l_{\text{personalized}}(\text{user},\text{interaction})]$, assuming homoscedasticity between $l_{\text{personalized}}(\text{user},\text{normal interaction})$ and $l_{\text{personalized}}(\text{user},\text{noisy interaction})$ for any user would result in equal variances for $l_{\text{overall}}(\text{normal interaction})$ 与 $l_{\text{overall}}(\text{noisy interaction})$, which does not align with the actual situation in Figures 1 and 2.
* **W2: selection of backbone is limited.** This paper only selects MF and LightGCN as backbone models, which are classic CF backbones. A more comprehensive backbone study, especially with more cutting-edge CF backbones, would better demonstrate the effectiveness of PLD.


### Question：

* **Q1: **Would it be better to swap the legend order in Figure 1, considering that the current placement of the legend is farther away from the corresponding bar chart? I initially misunderstood it.
* **Q2:** The results in Table 4 are under the BPR loss setup. It seems that the performance of many baselines is actually a  strongly negative effect. However, in Table 6, under the BCE loss setup, other baselines are effective. This result is quite strange. Why does the difference between BPR loss and BCE loss lead to such a significant discrepancy in the results?  It might be useful to explore the results on other datasets under the BCE loss setting.

If my concerns are addressed, I will be happy to raise my score.

**Reviewer Confidence:**

3: The reviewer is confident but not certain that the evaluation is correct

**Scope:**

4: The work is relevant to the Web and to the track, and is of broad interest to the community

---

### Official Review · Reviewer_m2fj · 2024-12-01

**Novelty:** 4
**Technical Quality:** 4

**Review:**

#### **Summary**

The paper addresses the challenge of noise in implicit feedback for recommender systems, caused by factors such as human error and behavioral ambiguity, which degrade recommendation accuracy and robustness. Existing methods fail to effectively distinguish between normal and noisy interactions due to significant overlap in the loss distribution, particularly with pairwise loss functions. To overcome this, the authors propose **PLD (Personal Loss Distribution)**, a resampling strategy that prioritizes normal interactions by leveraging a user’s personal loss distribution. Theoretical analysis and experiments on three datasets demonstrate PLD’s effectiveness and robustness across varying noise ratios.

#### **Pros**

- The paper is well-written, and the problem is clearly and convincingly motivated.
- The survey of related work is thorough and sufficient.
- The paper provides a fresh perspective by using the observation of a user's personal loss distribution to judge interaction noise.

#### **Cons**

- Reweighting strategies for interaction denoising in recommendation systems, based on predefined assumptions, have already been extensively studied. However, the authors fail to provide sufficient empirical evidence to support the assumptions proposed in this work.

- While the method section includes extensive theoretical analysis and discussion, it heavily relies on intuitive reasoning and unproven assumptions. The simplistic reweighting design and the lack of modeling for complex implicit feedback make it difficult to fully trust the method’s effectiveness.
- The paper is based on strong assumptions but lacks a detailed discussion of the proposed method's limitations and its applicability to specific usage scenarios.

**Questions:**

Using observations of users' personal loss distributions as a criterion for identifying interaction noise is a strong assumption. Can the authors provide empirical evidence across datasets in different application scenarios to validate the reliability of this foundational assumption?

**Reviewer Confidence:**

3: The reviewer is confident but not certain that the evaluation is correct

**Scope:**

4: The work is relevant to the Web and to the track, and is of broad interest to the community

---

### Official Review · Reviewer_vvZK · 2024-12-02

**Novelty:** 5
**Technical Quality:** 6

**Review:**

### Summary

This paper discusses identifying noisy samples based on training loss to reduce the impact of noisy samples during training. The authors observe that there is significant overlap between the loss of noisy and clean samples, making it difficult to distinguish between them based solely on the magnitude of the loss. Based on a synthetic experiment, they find that, from a user-level perspective, the loss of the two types of samples exhibits a greater difference. Based on this observation, the authors propose a user-level denoising method, providing theoretical analysis and experimental validation to support their approach.

### Pros

- The author provides some theoretical analysis of the proposed method.
- The author conducts a comprehensive set of experiments on the proposed method and theory.
- This paper is well-written and easy to follow.

### Cons

- **Insufficient Motivation**: The paper lacks an in-depth explanation of why the person-level loss has less overlap than the overall loss. Section 4.1 only discusses this phenomenon in a synthetic dataset (where the noise is independently and randomly generated), but there is a lack of further theoretical or intuitive explanation. Moreover, it remains unknown whether the noise in the real-world scenario follows such a characteristic.

- **Strong Assumption**: Theorem 1 assumes that the loss follows a Gaussian distribution, but there is no further explanation. Generally speaking, this assumption holds only when the loss is based on MSE, while the losses chosen in this paper are BPR and BCE. I understand that this assumption makes the calculations easier, but such a strong assumption undermines the generalizability of the theoretical conclusions.


---

# After the rebuttal

The authors claim that the Theorem 1 can disgard the strong Gaussian assumption. Additionally, the authors provide an intuitive explanation and interesting theoretical analysis to support the motivation. As most of my concerns are addressed, I decided to increase my scores from (4,4) to (5,6).

**Questions:**

See weaknesses. Additionally, I would like to know:

- Is it possible to provide a theorem for proving the person-level loss distribution has less overlap than the overall loss?

- Is it possible to remove the Gaussian assumption in Theorem 1? For example, only assume that the loss has a mean of $\mu$ and the variance $\sigma^2$, instead of assuming it follows $\mathcal N(\mu, \sigma^2)$?

**Reviewer Confidence:**

3: The reviewer is confident but not certain that the evaluation is correct

**Scope:**

4: The work is relevant to the Web and to the track, and is of broad interest to the community

---

### Official Review · Reviewer_YqFP · 2024-12-02

**Novelty:** 4
**Technical Quality:** 5

**Review:**

This paper addresses noise in implicit feedback for recommender systems, highlighting limitations in loss-based denoising due to overlap between normal and noisy interactions. The authors propose PLD (Personal Loss Distribution), a resampling strategy leveraging user-specific loss distributions to better distinguish interactions.

Pros:
1. This paper convincingly demonstrates, through data analysis, the limitations of traditional methods that address collaborative filtering noise by globally assigning loss weights.
2. The introduction of the PLD method, which resamples based on personal loss distributions, is a novel and potentially effective solution to the problem. The theoretical analysis supporting the method is also a strength.
3. Extensive experiments show PLD improves accuracy and robustness over existing methods.

Cons:
1. The paper lacks a case study with real user interactions to showcase the denoising effects before and after applying the method.
2. The theoretical conclusions presented in the paper could be more concise.

**Questions:**

Pls see the comments above.

**Reviewer Confidence:**

3: The reviewer is confident but not certain that the evaluation is correct

**Scope:**

4: The work is relevant to the Web and to the track, and is of broad interest to the community

---

### Official Review · Reviewer_EJ4t · 2024-12-02

**Novelty:** 3
**Technical Quality:** 4

**Review:**

The paper introduces a Personal Loss Distribution-based Denoising (PLD) strategy, which shifts the denoising focus from the overall loss distribution to the personal loss distributions of individual users. The method:

1.	Analyze users' personal loss distributions to distinguish noisy and normal interactions.
2.	Implements a resampling strategy where candidate interactions are prioritized based on their likelihood of being normal, reducing the probability of optimizing noisy interactions.

Pros

1.	Novel Approach:
The introduction of personal loss distributions for denoising interactions is innovative and addresses the limitations of global loss distributions effectively.

2.	Comprehensive Experiments:
Extensive evaluations on diverse datasets (e.g., Gowalla, Yelp2018, MIND) demonstrate the robustness and applicability of the proposed method under varying noise conditions.

3.	Theoretical Rigor:
The method is supported by a strong theoretical foundation, with mathematical proofs and analysis validating its effectiveness.

4.	Robustness to Noise:
The method’s robustness across different noise ratios and its consistent performance highlight its practical utility in noisy environments.

Cons:

1. The contribution of this work feels somewhat incremental. The biggest innovation lies in the shift from using global loss distributions, as in previous methods, to employing personal loss distributions. While this approach refines the existing methods, it doesn’t seem to represent a significant leap forward. In addition, what is the overlap ratio in the training? The R-CE and T-CE methods drop noisy samples at the very beginning of the training phase.

2.	Lack of Consideration for Noise Distribution Variations Across Users:

While the paper emphasizes personalized denoising, it fails to account for variations in noise levels across different users in its experimental design. All experiments assume a uniform noise ratio for all users, which contradicts the personalized premise of PLD.

This omission is particularly concerning given the results in Figure 5, where the performance of baseline methods like LightGCN is very close to PLD at various noise levels. The stability of the performance gap across noise levels suggests that a uniform noise ratio may disproportionately benefit PLD, undermining its real-world applicability.

3.	Limited Innovation Beyond Training Optimization:

The primary innovation of PLD lies in modifying the training process through personalized resampling, without making fundamental changes to the model architectures or addressing core challenges in representation learning.

The method remains reliant on backbone models like MF and LightGCN. This approach blurs the distinction between PLD and existing methods like DCF and self-supervised learning techniques, especially since these also involve data augmentation or weighting strategies for denoising.

In Figure 5, as the noise level increases, the performance of PLD converges to that of DCF, particularly with the MIND dataset and LightGCN as the backbone, raising concerns about the novelty and effectiveness of the proposed resampling strategy.

4.	Unjustified Addition of Temperature Coefficient $\tau$:

The paper introduces a temperature coefficient $\tau$ to the resampling probability $P_{u,v}$  without sufficiently justifying its impact on the theoretical formulation of $\xi$ or its influence on the practical implementation of Theorem 1. While $\tau$ is analyzed in the hyperparameter section, its inclusion lacks a clear mathematical or conceptual explanation in the methodology.

This omission weakens the theoretical foundation of PLD, leaving the reader questioning whether the performance improvements are due to better theoretical grounding or merely empirical tuning of hyperparameters.

5.	Baseline Comparisons Lack Cutting-Edge Methods:

The baseline methods selected for comparison (e.g., R-CE, T-CE, DCF, and DeCA) are standard but do not represent the most advanced or recent denoising techniques in recommendation systems. Incorporating more sophisticated approaches, such as those leveraging graph-based denoising or advanced self-supervised learning paradigms, would provide a stronger benchmark for evaluating PLD's efficacy.

**Questions:**

1.	Noise Distribution Variation Across Users:

How would the performance of PLD change if the noise level varied across users rather than being uniformly distributed?

Does the stability of PLD's performance gap with LightGCN across noise levels indicate that a uniform noise assumption is overly favorable to PLD?

Would introducing personalized noise levels better reflect the real-world applicability of PLD and align more closely with the personalized denoising concept?

2.	Innovation Scope:
Is PLD fundamentally different from data augmentation strategies like those in DCF or self-supervised learning methods?

Given that PLD relies on backbone models (MF, LightGCN), how does it address the limitations inherent to those models?

Does the convergence of PLD's performance with DCF at higher noise levels (e.g., on MIND with LightGCN) suggest that the resampling strategy adds minimal value beyond existing approaches?

3.	Temperature Coefficient $\tau$ Justification:

How does the introduction of $\tau$ alter the mathematical formulation of $\xi$ in the resampling probability $P_{u,v}$?

Can the authors provide a theoretical justification for how \(\tau\) interacts with other hyperparameters to influence denoising performance?

4.	Baseline Methods:

Why were cutting-edge denoising methods not included as baselines in the experiments?

How does PLD compare to the latest advancements in graph-based or self-supervised denoising techniques for recommendation systems?

Would the inclusion of these methods alter the conclusions about PLD’s efficacy?

**Reviewer Confidence:**

4: The reviewer is certain that the evaluation is correct and very familiar with the relevant literature

**Scope:**

3: The work is somewhat relevant to the Web and to the track, and is of narrow interest to a sub-community